# Comparing Viral Vectors and Fate Mapping Approaches for Astrocyte-to-Neuron Reprogramming in the Injured Mouse Cerebral Cortex

**DOI:** 10.3390/cells13171408

**Published:** 2024-08-23

**Authors:** Matteo Puglisi, Chu Lan Lao, Gulzar Wani, Giacomo Masserdotti, Riccardo Bocchi, Magdalena Götz

**Affiliations:** 1Division of Physiological Genomics, Biomedical Center, Ludwig-Maximilians-Universität München, 82152 Planegg-Martinsried, Germany; matteo.puglisi@bmc.med.lmu.de (M.P.); chulan.lao@med.uni-muenchen.de (C.L.L.); gulzar.wani@bmc.med.lmu.de (G.W.); giacomo.masserdotti@helmholtz-munich.de (G.M.); riccardo.bocchi@unige.ch (R.B.); 2Institute for Stem Cell Research, Helmholtz Zentrum München Deutsches Forschungszentrum für Gesundheit und Umwelt (GmbH), 85764 Nuremberg, Germany; 3Graduate School of Systemic Neuroscience, Biomedical Center, Ludwig-Maximilians-Universität München, 82152 Planegg-Martinsried, Germany; 4Munich Cluster for Systems Neurology (SyNergy), Biomedical Center, Ludwig-Maximilians-Universität München, 82152 Planegg-Martinsried, Germany

**Keywords:** astrocytes, neurons, direct reprogramming, fate mapping, birthdating, viral vectors, retrovirus, AAV, Neurogenin2

## Abstract

Direct neuronal reprogramming is a promising approach to replace neurons lost due to disease via the conversion of endogenous glia reacting to brain injury into neurons. However, it is essential to demonstrate that the newly generated neurons originate from glial cells and/or show that they are not pre-existing endogenous neurons. Here, we use controls for both requirements while comparing two viral vector systems (Mo-MLVs and AAVs) for the expression of the same neurogenic factor, the phosphorylation-resistant form of Neurogenin2. Our results show that Mo-MLVs targeting proliferating glial cells after traumatic brain injury reliably convert astrocytes into neurons, as assessed by genetic fate mapping of astrocytes. Conversely, expressing the same neurogenic factor in a flexed AAV system results in artefactual labelling of endogenous neurons fatemapped by birthdating in development that are negative for the genetic fate mapping marker induced in astrocytes. These results are further corroborated by chronic live in vivo imaging. Taken together, the phosphorylation-resistant form of Neurogenin2 is more efficient in reprogramming reactive glia into neurons than its wildtype counterpart in vivo using retroviral vectors (Mo-MLVs) targeting proliferating glia. Conversely, AAV-mediated expression generates artefacts and is not sufficient to achieve fate conversion.

## 1. Introduction

Many neurological diseases are still incurable, because the loss of neurons cannot be treated by neuronal replacement. However, converting local glial cells into neurons by the expression of a transcription factor that instructs neurogenesis could change this dire situation. This method has been pioneered first in vitro [1] and then in vivo [2] by expressing the neurogenic factor Pax6 specifically in proliferating glia using Moloney murine leukaemia retroviral vectors (Mo-MLVs). These viral vectors require the breakdown of the nuclear envelope to integrate their genome into the host cell DNA [3,4], and, hence, can stably integrate and transduce only cells undergoing mitosis. With this approach, other neurogenic factors, such as Neurogenin2 (Ngn2) and NeuroD1, proved to be more potent than Pax6 in reprogramming proliferative cortical glia into neurons in vivo [5,6,7]. However, Mo-MLVs and other retroviruses used for gene therapies induce strong inflammation when injected into the brain [8]. Ferroptosis and oxidative stress, which are detrimental effects of brain injury, are major hurdles in glial reprogramming, and their inhibition further increases direct conversion into neurons both in vitro and in vivo [6].

The lower immunogenicity and the lack of integration into the host genome make adeno-associated viruses (AAVs) more desirable vectors for human neurological gene therapies [9]. Since AAVs can promiscuously infect multiple postmitotic cell types, special strategies must be adopted to restrict their expression to astrocytes [10]. One strategy is to use cell-specific promoters to constrain the expression of the neurogenic factor to only one of the multiple infected cell types. For example, the expression of AAV vectors was thought to be limited to astrocytes by using elements of the Gfap promoter [11]. In direct reprogramming experiments, however, neurogenic factors can activate the Gfap promoter in cis, thus causing the expression to also occur in the infected neurons [12]. Indeed, earlier reports described that the Gfap promoter could be regulated by the different transgenes expressed downstream of it [13]. Since then, the original finding of astrocyte-to-neuron direct reprogramming using AAVs with Gfap regulatory elements driving neurogenic factors or knocking down/out Polypyrimidine Tract Binding Protein 1 (PTBP1) could not be reproduced [14,15,16].

The use of adequate controls is crucial to distinguish direct neuronal reprogramming, i.e., the conversion of glia into neurons, from the artefactual labelling of endogenous neurons [17]. This can be achieved by labelling the starter cells (e.g., astrocytes) to follow them over time, including their eventual conversion into neurons, or by labelling endogenous neurons to detect possible mis-expression of the viral transgene in them. Here we apply both strategies to compare in vivo direct reprogramming using AAV or Mo-MLV vectors with the same neurogenic factor. In addition, as further unbiased proof of in vivo direct reprogramming, we employ chronic in vivo live imaging aiming to observe the conversion of a cell with astrocytic morphology into a neuron [17]. 

As the neurogenic factor of choice, we used the more potent phospho-resistant form of the proneural factor Neurogenin2 containing the mutation of nine serine residues into alanine (9SA-Ngn2). 9SA-Ngn2 has increased neurogenic activity in development [18,19,20,21,22] and during in vitro direct conversion of astrocytes to neurons, revealing substantial improvements at all epigenetic levels [23]. We aimed here to determine if this factor would also be more effective than Ngn2 after a cortical stab wound (SW) injury in vivo. Ngn2 alone performs poorly, generating only about 10% of neurons and requiring the presence of co-factors [6,8]. Therefore, we tested how 9SA-Ngn2 on its own would perform in converting reactive astrocytes into neurons after a stab wound injury and which viral vectors would be best suited. 

## 2. Materials and Methods

### 2.1. Animals

Animal handling and experimental procedures were performed according to animal welfare policies and approved by the Government of Upper Bavaria (Germany). Both male and female animals were used for the experimental procedure. Mice were maintained in specified pathogen-free conditions at the Core Facility Animal Models, Biomedical Center (BMC), Faculty of Medicine, LMU Munich, in a 12:12 h light/dark cycle. Mice were housed in groups of 2–3 animals in individually ventilated cage systems in a room maintained at a temperature of 22 +/− 2 °C and 55 +/− 10% relative humidity. Mice had free access to water (acidified and desalinated) and standard rodent chow (Altromin 1310M, Altromin Spezialfutter, Lage, Germany). C57Bl6/J wildtypes (Jackson Laboratory #000664), GFAP::Cre (B6.Cg-Tg(Gfap-cre)77.6Mvs/2J, Jackson Laboratory #024098) [24], Aldh1l1::Cre (Tg(Aldh1l1-cre)JD1884Htz/J, Jackson Laboratory #023748) and GFPrep (FVB.B6-Tg(CAG-cat,-EGFP)1Rbns/KrnzJ, Jackson Laboratory #024636) [25] mouse lines were used in this study, including crosses to obtain GFAP::Cre/GFPrep mice. The genotyping for the GFAP::Cre transgene was performed using the primers Cre-Fw: (GCGGTCTGGCAGTAAAAACTATC) and Cre-Rv (GTGAAACAGCATTGCTGTCACTT), and for the GFPreporter using the primers AG2 (CTGCTAACCATGTTCATGCC) and CAT2 (GGTACATTGAGCAACTGACTG).

### 2.2. Labelling Endogenous Neurons with EdU

To label endogenous neurons, pregnant females were treated with a solution of 0.5 mg/mL 5-ethynyl-2′-deoxyuridine (EdU) (#E10187, Invitrogen, Waltham, MA, USA) and 1% sucrose (1-075651 Merck Millipore, Burlington, MA, USA) in drinking water between E7.5 and P0. The solution was changed every 2–3 days to avoid bacterial contamination.

### 2.3. Viral Vector Preparation

The plasmids pAAV-CAG-FLEX-mScarlet (Plasmid #99280, Addgene, Watertown, MA, USA) and pAAV-CAG-FLEX-9SA-Ngn2 were cloned via PCR cloning with SalI/BamHI in pAAV-CAG-FLEX-MCS vector (self-made). The RV plasmids pRV-CAG-9SA-Ngn2-IRES-mScarlet, pRV-CAG-9SA-Ngn2-IRES-dsRed2, and pRV-CAG-Bcl2-IRES-dsRed2 were cloned via Gateway cloning. In brief, 9SA-Ngn2 was synthesised using GenScript Biotech (Piscataway, NJ, USA) with a SalI/EcoRV restriction enzyme site, and sub-cloned into pENTR1A vector with IRES-mScarlert or IRES-dsRed2. All the rAAV2/5 vectors were produced and titred as described in [8], while Mo-MLVs were prepared as in [6]. For the AAV experiments, purified vectors were diluted in 1xPBS-MK buffer (1 mmol/L MgCl_2_, 2.5 mmol/L KCl, 0.001% Pluronic F68, 1× PBS) and mixed with 100 nL of Fast Green to reach the final concentrations of genome copies (gc)/mouse reported in Table 1, in a final volume of 500 nL/mouse. The MML-RVs purified mix was first used to infect mouse astrocytes cultured from postnatal day 5 (P5) C57BL6/J pups and calculate a functional titre in terms of transducing units (TU)/mL. Then, 5.8 × 10^6^ TU/mouse was mixed together with 100 nL of Fast Green to achieve a final volume of 500 nL and injected into the animals.

### 2.4. Cortical Stab Wound and Viral Injections

Adult mice (2–5 months old) were used for all experimental procedures. Animals destined for immunohistochemical analyses were subjected to cortical SWs and then viral injections three days after, as described in [6,8], with slight modifications. Briefly, mice were anaesthetised by intraperitoneal injection of a fentanyl (0.05 mg/kg; “Fentany-Priamal” Piramal Critical Care, Hallbergmoos, Germany), midazolam (5 mg/kg; “Dormicum” CHEPLAPHARM, Greifswald, Germany), and medetomidine (0.5 mg/kg; “Sededorm” Prodivet Pharmaceuticals, Eynatten, Belgium) mixture (MMF) and fixed in a stereotactic frame. Under a dissection microscope, a skin incision was used to access the skull and a high-speed microdrill was used to expose the caudal portion of the mouse motor cortex. A stab wound injury (SW) was then performed with a surgical blade at the following coordinates: −0.2 to −1.2 mm rostro-caudal, +1 mm medio-lateral, and −1 mm dorso-ventral. After that, the drilled bone flap was placed back and the skin was closed with surgical stitches. Three days after this, the injured area was re-exposed to perform the viral injections. All viruses were delivered in a final volume of 500 nL using pulled and bevelled glass capillaries connected to a nanoinjector set at a speed of 40 nL/min. Details about the viral mixes and titres are reported in Table 1. The glass capillary was left in place for 2 min before and after injection to increase viral diffusion and reduce flowback. Finally, anaesthesia was reversed using atipamezol (2.5 mg/kg; “Atipazole” Prodivet Pharmaceuticals, Eynatten, Belgium), flumazenil (0.5 mg/kg; “FLUMAzenil” B. Braun, Melsungen AG, Melsungen, Germany), and buprenorphine (0.1 mg/kg; “Buprenovet Multidose” Elanco GmbH, Cuxhaven, Germany) administered subcutaneously. 

Mice used for longitudinal in vivo imaging via 2-photon laser scanning microscopy (2PLSM) underwent a modified set of surgeries to accommodate the introduction of a cranial window. For all of them, surgeries were performed in a stereotactic apparatus provided with a heating pad to reduce animal distress, and the craniotomies were circular and 3 mm wide. Mice assigned to receive FLEX-AAV injections and 2PLSM underwent two surgeries. AAV-FLEX-GFP was injected during the craniotomy day, and CAG::FLEX-9SA-Ngn2 was injected fifteen days later. During the second surgery, a sterile 3 mm circular glass coverslip (#1 thickness, Warner Instruments, Hamden, CT, USA) was gently implanted into the craniotomy site and sealed in place with a thin layer of Sylgard (Sigma Aldrich, St. Louis, MO, USA) before dental cement (Dentalon plus, Heraeus Kulzer GmbH, Hanau, Germany) was applied. Finally, an aluminium chamber plate was fixed with cement on top of the glass coverslip to facilitate mouse head immobilisation at the 2-photon microscope via a head holder. Mice assigned to receive Mo-MLV injections and 2PLSM underwent similar surgical procedures to the immunohistochemistry ones, and the glass coverslips were implanted together with the aluminium chamber plates after the viral injection.

### 2.5. Immunohistochemistry

At the appropriate experimental timepoints, mice were anaesthetised and perfused trans-cardially with ice-cold phosphate buffered saline (PBS) first, followed by 4% paraformaldehyde diluted in PBS (4% PFA). Brains were post-fixed for 6 h in 4% PFA, then washed 3 times in PBS and either immediately processed or stored in a solution of 0.1% Sodium Azide prepared in PBS1×. Vibratome-cut 50–60 µm coronal brain sections were blocked in a solution of 3% Bovine Serum Albumin (BSA, A2153, Sigma-Aldrich) and 0.5% Triton ×100 (T8787, Sigma-Aldrich) diluted in 1 × PBS for >1 h at room temperature. Then, samples were incubated overnight at 4 °C with the appropriate primary antibodies and washed three times for 10 min in PBS the day after. If an EdU staining was performed on the same section, the primary antibody was fixed with an incubation of 10 min in 4% PFA before the EdU detection Click-IT reaction was performed according to the kit (C10340, Invitrogen). After three 10 min PBS washes, both EdU-stained and non-stained brain sections were incubated at room temperature for 2 h with the secondary antibodies and DAPI. Both primary and secondary antibodies were diluted in the blocking solution, and their specifics are reported in Table 2. Finally, brain sections were washed three times in PBS and mounted on microscope glass slides with Aqua Poly Mount (18606, Polysciences, Warrington, PA, USA) for imaging.

### 2.6. Imaging, Data Analysis, and Quantification

All images were acquired on a confocal laser scanning microscope, Zeiss LSM710 (Carl Zeiss Microscopy GmbH, Jena, Germany). Except for chronic live imaging, all quantifications were performed with at least three animals and three sections from each animal, as specified in the respective figure legend. Quantifications were performed using ImageJ software, and data were analysed with GraphPad Prism 7.0. Before proceeding with any statistical analysis, the normal distribution of the data was verified with the Shapiro–Wilk test. Depending on the normal distribution of the data, appropriate parametric or non-parametric tests were used, as specified in the legend of every figure. The bar plots in the figures were generated in GraphPad Prism and modified only for aesthetic purposes in Affinity Designer 1. Doughnut plots were generated directly in Affinity Designer 1. 

### 2.7. Chronic Live Imaging via 2-Photon Laser Scanning Microscopy (2PLSM)

In vivo imaging was performed using a Leica SP8 WLL DIVE FALCON microscope equipped with a 2-photon source (Insight X3 DUAL, Spectra-Physics, Milpitas, CA, USA). GFP and mScarlet were excited at 940 and 1160 nm, respectively, using a 16×, 0.8 NA, 3 mm working distance water immersion objective (Nikon, Tokyo, Japan). During the whole procedure, mice were under the same anaesthetic MMF mixture used for the surgeries. The mice were head-fixed using an aluminium headpost attached to a holder, which allowed consistent positioning of mice on the microscope stage, and maintained at 37 °C with a monitoring system (MARTA Pad, Vigilitec, Heiden, Switzerland). Imaging sessions lasted a maximum of 1 h and began 8 days post-second injection (dpi), continuing every second day until approximately 30 dpi. Anaesthesia was reversed using atipamezol (2.5 mg/kg; “Atipazole” Prodivet Pharmaceuticals), flumazenil (0.5 mg/kg; “FLUMAzenil” B. Braun, Melsungen AG), and buprenorphine (0.1 mg/kg; “Buprenovet Multidose” Elanco GmbH) administered subcutaneously.

## 3. Results

### 3.1. Fate Mapping of Cortical Astrocytes after AAV-FLEX-9SA-Ngn2 Administration 

First, we investigated whether 9SA-Ngn2 delivered via AAV could convert cortical astrocytes into neurons in a mouse model of traumatic brain injury. To do so, a cortical SW was performed as previously described [8] in Gfap::Cre/GFPrep mice (see Methods for details) to injure the cerebral cortex and induce glial reactivity. AAV vectors containing mScarlet-FLEX or 9SA-Ngn2-FLEX were injected either alone (mScarlet-FLEX) or in combination (mScarlet-FLEX and 9SA-Ngn2-FLEX) next to the injury site 3 days later (Figure 1A). FLEX refers to the gene orientation that is inverted and flanked by flex sites such that Cre recombinase is required to revert the orientation and allow gene expression [26]. Cortical astrocytes express *Gfap* both during postnatal development and after injury [27,28]. Therefore, our experimental system allows the simultaneous fate mapping of astrocytes via GFP labelling and the transduction of reactive astrocytes with FLEX-AAVs where the transgene should be expressed only in Cre-expressing cells due to its inversion. 

Four weeks after the injection, brains were analysed by immunostaining (Figure 1A). In the cortex of mice injected with 9SA-Ngn2 and mScarlet AAVs, we detected 22.3% NeuN+/mScarlet+ cells with a clear neuronal morphology (Figure 1B,C). On the contrary, only 3.3% of the mScarlet control AAV-infected cells were NeuN+ in the age-matched mice (Figure 1B,C). Such putative induced neurons (iNs; NeuN+ cells seen after 9SA-Ngn2 injection) were observed across all the cortical layers (Appendix A); however, only a few were also positive for GFP (NeuN+/mScarlet+/GFP+; Figure 1B,C and Appendix A), thus indicating a non-astrocytic origin of the NeuN+/mScarlet+ neurons. This result suggested that iNs may rather be due to artefactual activation of mScarlet in endogenous neurons. To examine this possibility, we further optimised the fate mapping of endogenous neurons. 

### 3.2. Fate Mapping of Endogenous Neurons after AAV-FLEX- 9SA-Ngn2 Administration

To interrogate the cellular origin of the NeuN+/mScarlet+ cells observed in the 9SA-Ngn2 mice, we fate mapped endogenous neurons using EdU. Nucleotide analogues, such as EdU, have been widely used to label neurons during the time of their generation in development [29,30,31], and this technique was recently employed to monitor endogenous neurons in astrocyte-to-neuron reprogramming [8]. Specifically, mice that were given EdU between embryonic day 7.5 and their birth (Figure 2A) were analysed at 1 month of age. Indeed, a clear nuclear EdU signal was observed in all the cortical layers (analysis focused on motor areas; Figure 2B–D and Appendix A). As expected, the vast majority of EdU+ cells were NeuN+ neurons (90.6% NeuN+/EdU). The protocol is highly specific and very efficient, since 97.3% of all the motor cortex neurons were labelled (Figure 2C and Appendix A). Only 0.9% of the EdU+ cells were Sox9+, a marker for astrocytes, which represented 3% of the astrocyte population in this cortical area (Figure 2C). This is the case because astrocyte progenitors proliferate at postnatal stages and hence dilute the EdU label. These data show that this EdU protocol allows labelling cortical neurons with high sensitivity and efficiency, and with a minimal leak in astrocytes. 

Next, Gfap::Cre/GFP pregnant females were similarly exposed to EdU between E7.5 and P0 to label the endogenous neurons of their progeny. The adult offspring were then subjected to SW injury and, 3 days later, injected with FLEX-AAV vectors carrying either the control or the 9SA-Ngn2 construct (Figure 2E and Appendix A). Analysis was performed 4 weeks after injection, as described above. Also, in this case, the EdU labelling was present in the whole cerebral cortex grey matter (GM) and it was more intense in the deep layers (Figure 2F,G and Appendix A). Putative iNs were observed in all cortical layers, albeit with a lower proportion than before (around 8.6% NeuN+/mScarlet+ cells; Figure 2H and Appendix A). This may be due to some degree of toxicity of EdU (see Discussion). When inspected for EdU labelling, most NeuN+/mScarlet+ cells had an EdU+ nucleus (97.6% EdU+/NeuN+/mScarlet+), suggesting that almost all of the alleged iNs are endogenous neurons. Notably, 16.9% of the Sox9+/mScarlet+ glial cells (78.4% of all mScarlet+) also had an EdU+ nucleus. This higher percentage compared to the intact brains may be due to up-take of EdU from dying neurons, when astrocytes are triggered to proliferate after the injury ([32,33]). 

### 3.3. Longitudinal In Vivo Imaging of FLEX-AAV-Labelled Cells

To examine the behaviour of cells transduced by the FLEX-AAVs, we performed longitudinal in vivo imaging of the mouse cortex by 2PLSM from 8 days after the second viral vector injection (Figure 3A,B). As the onset of fluorescence driven by the AAV constructs is very slow, we injected the AAVs with the fluorescent reporter 15 days prior to the window implantation and 9SA-Ngn2FLEX injection (Figure 3A). Cells imaged over weeks (Figure 3B) were classified as astrocytes (green), neurons (red), or unknown (grey) according to their morphology (Figure 3C). We observed bushy astrocytes and many neurons (Figure 3D), but rarely intermediates. Most cells persisted over the experiment and maintained their morphological identity as astrocytes or neurons. The example in Figure 3D may be one of the cases where a transition in morphology occurs; however, even in this example the neuronal-looking cell with a clear soma visible at day 19 may not have been derived from the astrocyte, but rather appeared de novo, like many other neurons. Indeed, to better prove conversion from astrocytes, it would be important to combine imaging with an astrocyte reporter. As we could not find any more unequivocal example of a cell with a glial morphology converging into a neuronal one, chronic live in vivo imaging could not demonstrate the conversion of 9SA-Ngn2 FLEX-AAV-transduced astrocytes to neurons. Conversely, many cases of de novo appearance of neurons were observed, consistent with the up-regulation of the reporter in endogenous neurons. 

### 3.4. Control of FLEX-AAV

These data suggest that FLEX-AAVs may be artefactually expressed in endogenous neurons. Off-target expression of FLEX-AAVs may depend on low-level expression of the inverted transgene or recombination during plasmid production. In both scenarios, transgenes coded by FLEX-AAVs are expressed in a Cre-independent manner [34]. To investigate whether this is also happening in our case, we repeated the previous experiment in Cre-negative mice (Appendix A). The injection in Cre-negative mice coming from GFAP::Cre and Aldh1l1::Cre litters suggested that such events cannot explain the results above. In fact, half of the analysed brain sections showed no labelled cells at all, and the other half contained just one neuron (Appendix A). Thus, we can exclude Cre-independent expression of our vectors as cause for the expression in endogenous neurons. 

### 3.5. Intracerebral Reprogramming with 9SA-Ngn2 Using Mo-MLVs

To probe if 9SA-Ngn2 can reprogram proliferating reactive glial cells, we used the Mo-MLV retroviral vectors, as they integrate and are stably expressed only in proliferating cells, such as reactive astrocytes and oligodendrocyte progenitor cells (OPCs). Moreover, they have already been used to reprogram adult astrocytes into neurons in vivo ([5,6,7,35]). Notably, when wildtype Ngn2 was delivered to proliferating reactive cortical glia via Mo-MLVs, less than 10% of the targeted cells were NeuN+ [6]. To test how 9SA-Ngn2 would reprogram proliferating reactive glial cells in vivo, we injected Mo-MLV-CAG::9SA-Ngn2-IRES-mScarlet into the brains of adult Gfap::Cre/GFPrep mice three days after SW. Analysis was performed via immunohistochemistry at 10 and 28 days post-injection (dpi; Figure 4A), the same timepoints used for wildtype Ngn2 [6]. NeuN+/mScarlet+ cells were observed at both timepoints, increasing from 36% amongst all mScarlet+ cells at 10 dpi to 63.8% at 28 dpi (Figure 4B–D). Notably, 88.6% and 76% of these NeuN+/mScarlet+ cells were also GFP+ at 10 dpi and 28 dpi, respectively. These data suggest that most of the NeuN+/mScarlet+ cells observed in our experiment originate from astrocytes (Figure 4B–E). We also analysed the morphology of these cells classified as bipolar, multipolar, or round with no detectable processes, as shown in Appendix A. Most of the GFP+ iNs had a multipolar morphology, while most of the very few GFP-negative iNs had a bipolar morphology, even though this difference was not significant. The latter also comprised more cells with a round morphology. This may be due to persistence of the morphology from the starter cells—more multipolar reactive astrocytes and bipolar reactive oligodendrocyte progenitors—or a slightly better transition to a bipolar neuronal morphology amongst the GFP-negative cells. Importantly, we could observe some cells with clear neuronal features, such as bipolar morphology and/or extending long axon-like processes (Figure 4B,C and Appendix A). Taken together, Mo-MLVs expressing 9SA-Ngn2 succeed in instructing the conversion of endogenous reactive proliferating astrocytes into iNs with an outcome exceeding the capacity of wildtype Ngn2. 

## 4. Discussion

The present study aimed at addressing a fundamental question in the field of in vivo direct reprogramming, namely the origin of iNs and possible pitfalls depending on the viral delivery method employed. Towards this goal, we showed an efficient birthdating method to label endogenous neurons and employed genetic fate mapping of astrocytes, while applying the same reprogramming factor in two different viral vector systems. We expressed the phospho-resistant form of Ngn2 and found that it is also more efficient than wildtype Nng2 in in vivo direct reprogramming, as it was recently shown to be the case in vitro [23]. 

### 4.1. Reprogramming via Mo-MLVs Is Reliable and 9SA-Ngn2 Is More Potent Than Its Wildtype Counterpart

In line with the much more effective astrocyte-to-neuron conversion mediated by 9SA-Ngn2 compared to Ngn2 in vitro ([23,36]), we found that 9SA-Ngn2 was up to 5x more efficient than its wildtype counterpart in generating new neurons in vivo [6]. Mo-MLV-based retroviral vectors should not target postmitotic neurons, as breakdown of the nuclear envelope is required for their integration into the host genome. However, we further ensured their reliability by genetic fate mapping. Genetic fate mapping of astrocytes using GFAP::Cre/GFPrep mice showed that the majority of the newly generated neurons (or iNs) are indeed derived from astrocytes. Interestingly, we could also observe a minority of NeuN+/mScarlet+ cells that were GFP-negative. Given that the other proliferating cell type targeted by the Mo-MLVs consists of proliferating OPCs [5], we hypothesise that they could be the source of these few GFP-negative iNs. Interestingly, such iNs persist and may even survive better than the astrocyte-derived iNs, but their extremely limited number and variability make it difficult to draw any firm conclusion (Appendix A). Ideally, survival is best examined by live imaging, which has revealed a large contribution of cell death when following cells during neuronal reprogramming in vitro [6]. In vivo live imaging proved to be extremely difficult in the context of the inflammatory environment elicited by the SW and Mo-MLV injection. We could observe some cells undergoing morphological changes reminiscent of neuronal conversion, but, as the imaging quality was limited, we refrain from any conclusions from these experiments. However, when using Ngn2 and Bcl2 as a gold standard for Mo-MLV-mediated reprogramming, we could observe better examples. Likewise, using the allegedly also more efficient NeuroD1 transcription factor for reprogramming, many iNs were also observed by live imaging, even though they were not traced back individually to a glial cell origin [37]. Indeed, it is important to note that 9SA-Ngn2-mediated reprogramming via Mo-MLVs is significantly less efficient and results in less mature iNs compared with the Mo-MLV-delivered combination of wildtype Ngn2 and Bcl2 [6]. Neurons induced with 9SA-Ngn2 did not develop such a mature morphology as achieved in the combination with Bcl2 or Nurr1 [6,8]. Taken together, 9SA-Ngn2 achieves a better reprogramming outcome than wildtype Ngn2 in vivo as well, supposedly by the same molecular mechanisms as recently unravelled in vitro, namely more efficient opening of chromatin and expression of neurogenic genes as well as improved chromatin loop formation and DNA demethylation [23]. However, combination with other factors, e.g., Bcl2, is needed to achieve better survival and more complete maturation. 

### 4.2. Expression of Phospho-Resistant Ngn2 by AAV Labels Endogenous Neurons and Does Not Result in Direct Reprogramming

Given these results, we were also expecting an improvement in the reprogramming efficiency when switching to AAV vectors, because they elicit less inflammation than Mo-MLVs [8]. However, all neurons detected after the injection of AAVs containing 9SA-Ngn2 were labelled by EdU, which was used to mark endogenous neurons during embryonic neurogenesis. In line with this, the same cells targeted with AAVs containing 9SA-Ngn2 were negative for GFP, the fluorescent protein used to fate map astrocytes during the experiment. These converging lines of evidence support the conclusion that alleged iNs are indeed not derived from astrocytes, but rather are mostly endogenous neurons that artefactually express mScarlet. Finally, we also performed chronic live imaging after FLEX-AAV delivery. Consistent with previous results, most neurons maintained their morphology during the whole experiment and only 1–2 examples were possibly coming from cells with an astrocyte morphology. Admittedly, however, these imaging data have some limitations. The live imaging was performed in mice without a cortical stab wound injury. While this reduces the effect of brain inflammation on the quality of imaging, it also reduces astrocyte reactivity and possibly affects reprogramming [8]. Moreover, no astrocyte reporter was included in these experiments. 

This prompts two interesting questions. First, how is this artefactual expression in neurons brought about, and second, why does the AAV-driven 9SA-Ngn2 not reprogram almost any astrocyte? Regarding the first question, Wang et al., 2021 described the artefactual expression occurring in endogenous neurons only, when the viral vector also expresses the neurogenic reprogramming factor, but not in cells expressing only the fluorescent protein. This excludes the possibility of artefactual Cre expression in neurons, e.g., elicited by the stab wound injury, under control of the GFAP promoter (for debate on Gfap expression in neurons after injury, see [38,39,40], as this should then also lead to expression in neurons using the control virus and the expression of Cre-dependent lineage trace reporters in iNs. 

Wang and others have shown that using the 681 bp Gfap promoter (GfaABC1D) to restrict the expression of neurogenic factors to astrocytes was not reliable in a viral vector [12]. Indeed, previous evidence suggested that this promoter is influenced by the transgenes downstream, leading to expression in both astrocytes and neurons [13]. Different strategies to post-translationally silence the GfaABC1D-driven transgenes’ expression in neurons exist and have been successfully applied to selectively transduce astrocytes [41,42,43]. Notably, while Wang et al. observed the leaky expression of GfaABC1D -AAVs vectors in endogenous neurons, the application of FLEX-AAVs came with an opposite conclusion. When FLEX-AAVs had been injected into the intact or injured mouse cortex of Aldh1l::CreERT2 mice, transgenes were specifically expressed in astrocytes, but none were converted into neurons [12]. These data suggest that injecting FLEX-AAVs into astrocyte–Cre-expressing mice should be a safer and more reliable option than the currently available “astrocyte-specific” AAVs. This is why we chose a transgenic mouse line using a longer regulatory element to drive Cre expression and found reliable astrocyte-specific expression of the control vector, but not the 9SA-Ngn2. We also showed that the expression of FLEX-AAVs does not occur in endogenous neurons in absence of the Cre. This may happen in rare cases depending on both ITR-mediated transcription and the presence of recombined rAAVs in the viral prep [34,44]. However, this was not the case in our experiments, as only single isolated neurons were found in a very small proportion of all the brain sections from injections into Cre-negative C57Bl6/J mice. As the Gfap promoter and 9SA-Ngn2 transgene were separated in our experiments, the in cis activation mechanism proposed by Wang et al. does not apply to our case. Moreover, wildtype Ngn2 is known to have a repressive rather than activating effect on the Gfap promoter [45]. 

Therefore, mechanisms independent from the transcriptional interaction between 9SA-Ngn2 and Gfap should be considered. For example, the transfer of Cre proteins or mScarlet (both as protein and as episomal AAV genomes) from the originally targeted astrocytes to the endogenous neurons could be promoted by 9SA-Ngn2 either directly and/or indirectly. Upon Ngn2 expression, many target genes become activated, which causes ER stress [36]. This may promote the formation of extracellular vesicles, which have already been described as transporting Cre, or the formation of structures such as tunnelling nanotubes [46,47,48,49]. If the transfer of the Cre is the culprit of the artefact, limiting the recombination to the first days of reprogramming using inducible mouse lines such as Aldh1l1::CreERT2 may prevent the mis-targeting of endogenous neurons [50]. Even if the separation of 9SA-Ngn2 and mScarlet in two distinct AAV vectors prevents any in cis artefact generation, it is also a limitation of our experimental design. In our experiments, mice injected with the control viral mix received fewer AAVs, namely only mScarlet containing AAVs, than the mice injected with 2 AAVs (Scarlet; 9SA-Ngn2) in the reprogramming experiments. However, this was controlled for in previous experiments using two fluorescent proteins for control experiments [26]. Finally, the expression of the proneural factor in astrocytes could increase or change exosome release, causing recombination in endogenous neurons.

Beyond the artefactual expression of 9SA-Ngn2 and mScarlet in endogenous neurons, however, we would also expect some astrocytes to convert to neurons, as they also express 9SA-Ngn2. However, we could hardly detect any neurons converting from astrocytes, as observed by genetic fate mapping or live imaging. This suggests that the expression of 9SA-Ngn2 alone via the AAV is not sufficient to reprogram cortical astrocytes. This may be due to the slow and gradual onset of the expression and/or the expression levels, even if the same constitutive CAG promoter was used for both Mo-MLVs and AAVs. Notably, previous experiments combined FLEX Ngn2 and FLEX Nurr1 and observed many neurons that were not endogenous neurons according to the EdU labelling paradigm [8]. Thus, it may be that 9SA-Ngn2 on its own is not sufficient, when expressed by this slow and gradual-onset virus. However, this difference may also come from targeting only proliferating cells via the RV, while mostly postmitotic cells are targeted via the AAV. Indeed, proliferation has been shown to ease transcriptional rewiring in reprogramming [51], and proliferating reactive astrocytes appear to be particularly plastic [52]. Thus, several factors may contribute to the low to absent reprogramming seen with AAV-mediated delivery. Given the much smaller inflammatory reaction elicited by the AAV versus RV delivery, it is important to further optimise AAVs for use in direct reprogramming, ideally targeting the AAV to specific cell types and employing high expression strategies [9]. 

### 4.3. Technical Consideration of the Fate Mapping Controls

Here we used several methods to label specific cell types. One involved using a fluorescent reporter expressed under a ubiquitous promoter upon Cre-mediated excision in astrocytes. However, reporter gene expression is also weaker in some cells than others, and in some cases, we observed lower reporter gene expression in the lesion area. Therefore, it is advantageous to use several approaches to ensure reliability of the fate conversion. Here we used as a second method labelling of endogenous neurons by applying EdU during their birth. However, we noted a lower proportion of NeuN+/Scarlet+ cells after the delivery of the FLEX-AAVs in Gfap::Cre/GFPrep treated with EdU. Indeed, it has been proposed that nucleotide analogues may inhibit astrocytes’ reprogramming when provided immediately before the reprogramming process [53] to label proliferating astrocytes. In our experiments, however, EdU was provided a long time before viral injection, to label the endogenous neurons when they were generated during embryogenesis. Nevertheless, some EdU may be released during the death of neurons after the SW injury and then become incorporated within proliferating astrocytes, as is suggested by the increase in EdU-labelled astrocytes in the reprogrammed mice compared to intact ones (Figure 2A,F). Therefore, the impact of EdU on astrocyte reprogramming is also important to consider in this paradigm. However, the labelling of proliferating astrocytes still seems to allow reprogramming, as seen in the work from our and other labs [8,54,55,56]. 

Finally, we used in vivo live imaging to follow individual astrocytes during their conversion process. While this approach is of course the most direct proof for cell conversion, it can be further improved, when combined with the fate mapping of the starting cells or reporters for the emerging neurons (e.g., using a synapsin promoter-driven fluorescent protein). However, its disadvantage is that the chronic window implanted in the skull elicits further inflammation, thereby probably also resulting in underestimation of the true conversion rate. 

Taken together, despite the limitations that each of these techniques have, one or two complementary controls are required to validate the true glial origin of induced neurons and exclude the artefactual labelling of endogenous neurons. Since fully cell type-specific viral vectors do not exist yet, it is furthermore important to determine the consequences of neurogenic factor expression in postmitotic neurons. 

## Figures and Tables

**Figure 1 cells-13-01408-f001:**
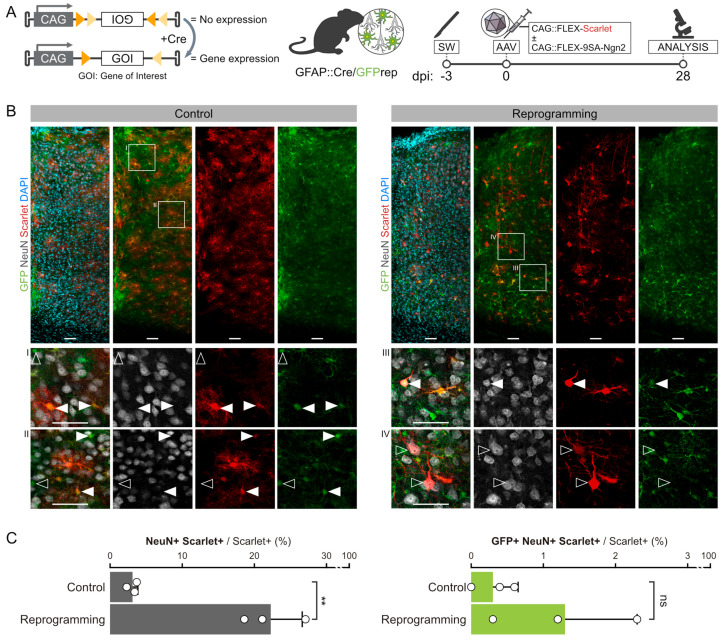
Genetic lineage tracing of endogenous astrocytes during FLEX-AAV transduction. (**A**) Schematic of the Cre-dependent FLEX-switch (left) and surgical procedures used to lineage trace endogenous astrocytes in SW-injured GFAP::Cre/GFPrep mice after the injection of FLEX-AAVs (right). (**B**) Confocal pictures of a 15 μm stack of the 28 dpi SW-injured motor cortex of GFAP::Cre/GFPrep mice injected with control (left, light grey) or 9SA-Ngn2 (right, dark grey) viral mixes. Single optical plane magnifications of a representative group of cells are reported in inserts I-IV. AAV-transduced cells (red) were tested for the expression of the lineage tracing reporter GFP (green) and neuronal marker NeuN (white). For control experiments, GFP+ astrocytes targeted by the FLEX-AAVs are pointed out by full arrowheads, while non-infected astrocytes are pointed out by empty arrowheads (inserts I and II). For reprogramming experiments, one of the rare astrocyte-derived GFP+/NeuN+/mScarlet+ cells is pointed out by a full arrowhead in insert III, while GFP-/NeuN+/mScarlet+ endogenous neurons are pointed out by empty arrowheads in inserts III and IV. Scalebars: 50 μm. (**C**) Quantification and lineage tracing analysis of the NeuN+/mScarlet+ cells belonging to the control or 9SA-Ngn2 experimental groups. mScarlet+ cells in all cortical layers that were less than 340 μm distant from the SW were included in the quantification. Every dot represents the values from one animal, for which three sections have been averaged. Statistical differences in the numbers of NeuN+/mScarlet+ or GFP+/NeuN+/mScarlet+ cells between control and SA-Ngn2 mice were calculated by an unpaired *t*-test. **: *p*-value < 0.01; ns: no statistically relevant difference.

**Figure 2 cells-13-01408-f002:**
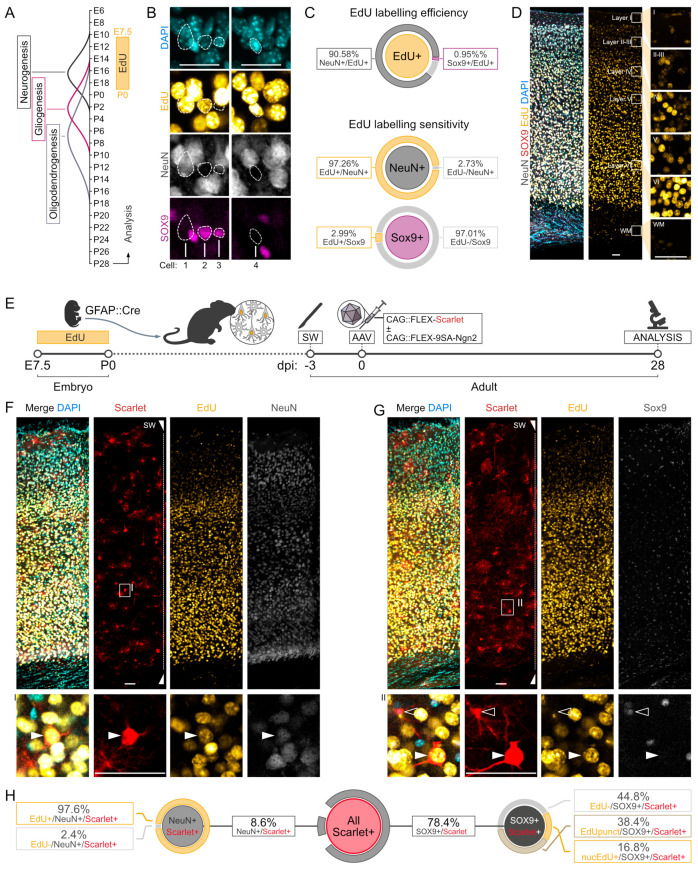
Chemical lineage tracing of endogenous neurons during FLEX-AAV transduction. (**A**) Timeline of the generation of the main cell types in the cortex: neurons (E12-E18), astrocytes (E18-P8), and oligodendrocytes (E18-P14). EdU was administered via drinking water between E7 and P0, and cell labelling in the motor cortex was analysed in 28-day-old mice. (**B**) Representative pictures of EdU-labelled cells in the mouse motor cortex. Astrocytes are marked by Sox9 and neurons by NeuN. Most of the endogenous neurons are labelled by EdU as Cell1, while astrocytes are not like Cell3. EdU rarely labels the nuclei of Sox9+ astrocytes as Cell2, or other unidentified cells as Cell4. Scalebar: 25 μm. (**C**) Quantification of EdU labelling specificity (top) and efficiency of neurons (middle) and astrocytes (bottom). Graphs show the fraction of cells positive for the marker at the centre of each circle that co-express one or more other markers. Exact mean ± SD values for each class were calculated from three different mice (one brain section each) and are reported on the side. (**D**) Confocal picture showing the pattern of EdU labelling across the cortical layers of the mouse motor cortex. Layer-specific magnifications are reported in the relative inserts. Scalebar: 50 μm. (**E**) Scheme illustrating the strategy used to pre-label endogenous neurons with EdU in adult GFAP::Cre mice that were subjected to cortical injury and viral injections. The two surgical procedures were 3 days apart, and mice were sacrificed 28 days after the AAV injection for the analysis. (**F**,**G**) Confocal pictures of the mouse motor cortex next to the stab wound injury (SW) 28 days after the AAV injection. Transduced cells are labelled by mScarlet (red) and were co-stained with anti-NeuN (grey in **F**) and anti-Sox9 (grey in **G**) to determine their identity, and with Alexa647 via Click chemistry to check the EdU labelling. A single optical plane magnification of an EdU+ neuron is reported in inserts I and II. Scalebars: 50 μm. (**H**) Quantification of the different cell groups observed in the experiment described in (**E**–**G**). mScarlet+ cells were checked for the independent expression of the neuronal marker NeuN and the astrocytic marker Sox9. After that, cells were examined for the presence of EdU. When present in NeuN+ neurons, the EdU signal was only nuclear. Conversely, when Sox9+ astrocytes had an EdU signal, this could have been both nuclear or within somatic puncta. The data in H represent the average of three brain sections coming from the same mouse.

**Figure 3 cells-13-01408-f003:**
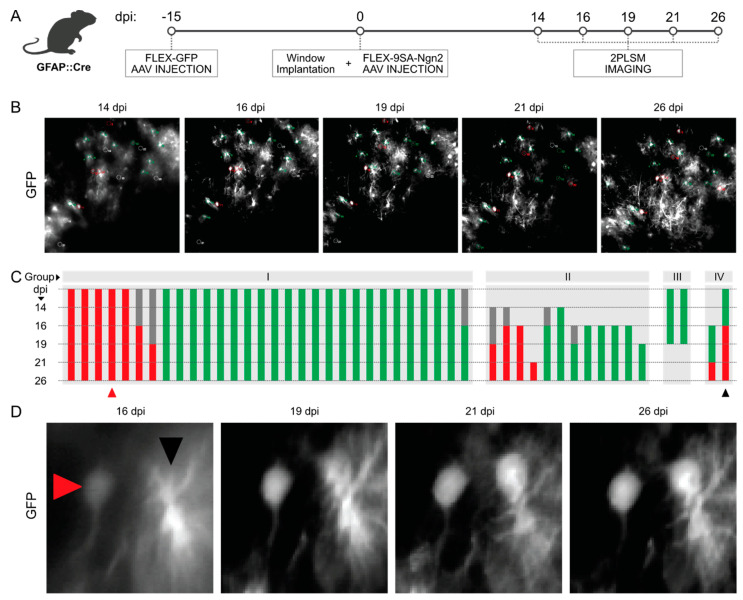
Chronic in vivo imaging of FLEX-AAVs via 2-photon microscopy. (**A**) Scheme illustrating the timeline of the experimental procedures used for the in vivo imaging of GFAP::Cre mice injected with FLEX-AAVs. (**B**) Pictures of the imaged cortical area at different timepoints (14, 16, 19, 21, 26 dpi). (**C**) Classification of the cells observed during the chronic in vivo imaging according to their morphology and dynamic during the experiments. Astrocytes (green) and neurons (red) have been divided into four different groups: cells observed since the earliest timepoint (I), cells observed only from later timepoints (II), cells that disappeared before the completion of the experiment (III), and cells that underwent a morphological change (IV). Examples of non-converting and one of the few possibly converting cells are indicated with a red and a black triangle, respectively, in (**D**).

**Figure 4 cells-13-01408-f004:**
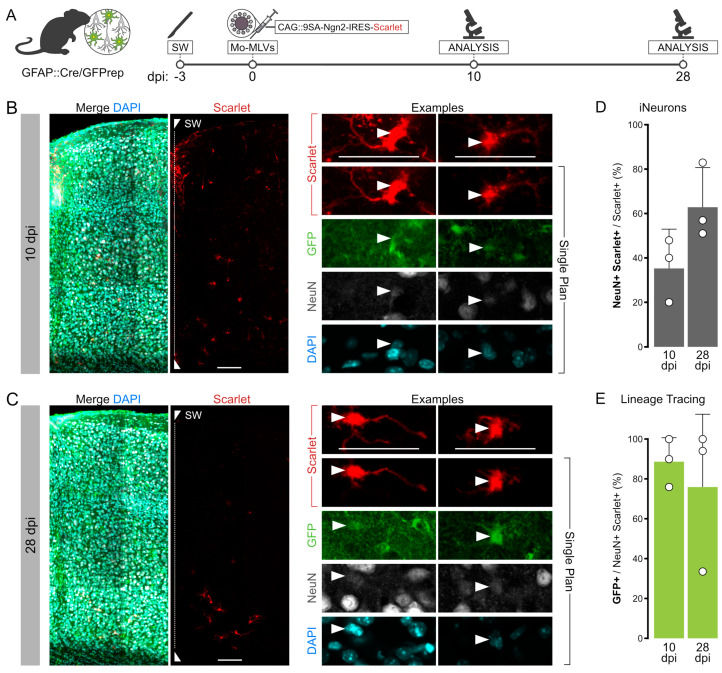
Genetic lineage tracing of endogenous astrocytes during Mo-MLVs transduction. (**A**) Schematic of the surgical procedures used to lineage trace endogenous astrocytes in SW-injured GFAP::Cre/GFPrep mice after the injection of Mo-MLVs. (**B**,**C**) Confocal pictures of a 15 μm stack of SW-injured motor cortex of GFAP::Cre/GFPrep mice at 10 (B) or 28 (**C**) days after the injection of Mo-MLVs-CAG::9SA-Ngn2-IRES-mScarlet. Scalebar 50 μm. Single optical plane magnifications of a representative group of cells are reported in inserts I-IV. Mo-MLVs-transduced cells (red) were double-stained for the lineage tracing reporter GFP (green) and neuronal marker NeuN (white). Scalebars: 50 μm. (**D**,**E**) Quantification and lineage tracing analysis of the NeuN+/mScarlet+ cells. Plotted data represent the mean ± SD of three biological and technical replicates.

**Table 1 cells-13-01408-t001:** Viral mixes and titres.

Virus Type	Mix	Virus Name	Virus Titre	Final Volume
AAVs	Control	CAG:FLEX-mScarlet or GFP	3.12 × 10^10^ gc/mouse	500 nL
	Reprogramming	CAG:FLEX-mScarlet or GFP	3.12 × 10^10^ gc/mouse	500 nL
		CAG:FLEX-9SA-Ngn2	4.90 × 10^11^ gc/mouse	
Mo-MLVs	Reprogramming	CAG-9SA-Ngn2-IRES-mScarlet	5.8 × 10^6^ TU/mouse	500 nL

**Table 2 cells-13-01408-t002:** Antibodies and DAPI.

Antibody/DAPI	Species/Isotype	Source	Identifier	Dilution
Anti-GFP	Chicken	AvesLab	GFP-1020	1:1000
Anti-RFP	Rabbit	VWR	ROCK600-401-379S	1:1000
Anti-NeuN	Mouse-IgG1	Merck Millipore	MAB377	1:250
Anti-Sox9	Rabbit	Merck Millipore	ab5535	1:1500
Anti-GFAP	Rabbit	Dako	Z0334	1:500
Anti-chicken-488	Donkey	Dianova	703-545-155	1:1000
Anti-rabbit-598	Donkey	Thermo Fisher	A-21207	1:1000
Anti-mouse IgG1-647	Goat	Thermo Fisher	A-21240	1:1000
DAPI	-	Sigma	28718-90-3	1:1000

## Data Availability

Data are available at https://doi.org/10.5281/zenodo.13359372.

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
