# Peer review of "Comparing Viral Vectors and Fate Mapping Approaches for Astrocyte-to-Neuron Reprogramming in the Injured Mouse Cerebral Cortex"

_cells, 2024, doi:10.3390/cells13171408_

Round 1

Reviewer 1 Report

Comments and Suggestions for Authors

Puglisi and colleagues report on their efforts to carefully characterize the performance of FLEX-AAV-based approaches to in vivo Neurog2-driven glia-to-neuron reprogramming, using vectors encoding for a “phosphomutant” form of Neurog2 (9SA-Neurog2). The authors employ a stab wound-based paradigm in the adult cortex of GFAP::Cre mice. Experiments performing either astroglial lineage-tracing using GFPreporter mice or extensive embryonic labeling of cortical neurons using EdU, both support the notion that most, if not all, AAV-labeled neurons are actually endogenous. The authors also report differential labelling of endogenous neurons by AAV-encoded Scarlet using control or Neurog2-encoding AAVs and, using Cre-negative littermates from GFAP::Cre and Adlh1l1::Cre mice, go on to demonstrate that, whatever the reason for this difference, Cre expression is required for endogenous neuronal labelling by FLEX-AAVs. In line with EdU-based analyses, live imaging of AAV-targeted glial cells provides no support for actual reprogramming (with mentioned caveats due to lack of injury). Finally, the authors tested 9SA-Neurog2 ability to drive in vivo reprogramming using retroviral targeting of proliferating glia in GFAP::Cre/GFPrep stab-wounded cortices, with potentially interesting observations regarding reprogramming cell population dynamics.

 All in all, this report provides solid and valuable evidence for the identity of FLEX-AAV-labeled neurons and clarifies the role of some elements (e.g. Cre-dependence), which should prove useful in future attempts by the field to characterize and possibly improve the behaviour of AAV-based systems. 

Comments

The study provide further compelling evidence that current AAV-based approaches to in vivo reprogramming are mostly not only ineffective at converting glia, but are also complicated by unspecific labeling of endogenous neurons.

·       The authors refer in passing a few times to their own report of AAV-based in vivo reprogramming (Mattugini et al. 2019), which differed in certain experimental aspects (such as a much shorter duration of EdU-based embryonic labeling of endogenous neurons and different reprogramming factors employed) from the currently reported approach. It is unclear if and how such experimental differences may account for the different interpretations of the results in these two reports. Given how the current report adds to an important and growing list of cautionary tales regarding the use of AAVs for in vivo reprogramming, it would be important if it could also comprise a (however brief) discussion of how earlier observations may still be best explained as genuine examples of AAV-driven reprogramming or if, in light of observations reported here, extending current experimental controls to earlier reprogramming paradigms may be necessary.

·       Connected to the above, an important experiment to support actual reprogramming in Mattugini et al., had consisted of the post-lesional labelling of proliferating glia with EdU. Since the authors have now shown that, when using Neurog2 FLEX-AAVs in GFAP::Cre mice, almost all NeuN+/Scarlet+ cells are endogenous neurons, it may be useful to know what, if any, fraction of the NeuN+/Scarlet+ population may be labeled by post-lesional administration of EdU. The current report´s main value lays in the care placed in characterizing the employed AAV-based paradigm. This model would therefore be well-suited to provide a measure of "background" labeling by post-lesional nucleotide analog incorporation in a setting where such labeling should be negligible (given the endogenous nature of the neurons labeled by the FLEX-AAV system, here).

·       The authors report that a progressively increasing fraction (though a minority) of the NeuN+/Scarlet+ population in Mo-MLV-targeted cortices is GFP-negative. While the authors suggests this may reflect oligodendroglia-derived iNs surviving better or taking longer to acquire NeuN expression, low expression and gradual accumulation of Scarlet by infected and not transduced endogenous neurons may also be a reason. Is Scarlet expression visibly lower in GFP-negative putative iNs, or is their morphology more mature-looking than that of GFP+ neurons?

·       Referring to the above, and considering the possibility that GFP- NeuN+/Scarlet+ cells may indeed be induced neurons: it would be interesting to check if Dcx expression by GFP+ and GFP- Scarlet+ populations show distinct trends between 10 and 28dpi. Delayed maturation would predict a lower slope of Dcx expression decline among GFP- cells. In any case, differential survival, if relevant here, may also be affected by different maturation rates/states, making independent testing of the two hypotheses complicated.

·       The authors report Scarlet+ endogenous neurons as being found across all cortical layers, but do not comment on the relative abundance at different apicobasal locations. 

Minor points

·       The proposed example of imaging-tracked glial conversion to a neuron is, as the authors admit, not very convincing; indeed the rounded neuronal soma of Cell17 seems to be weakly visible by 16dpi already, a bit higher to the left of the position where the white arrowhead has been placed (by the way, the arrowhead color code seems to have been swapped between panels C and D).

·       It is not clear from the discussion in the text, what are the numbers of cells imaged in the context Mo-MLV-based reprogramming. The text is ambiguous as it seems to suggest multiple examples of glia-to-neuron morphological conversion were observed (line 402), to then refer to a single cell as developing a neuronal morphology by 17 dpi (line 403).

·       A few references are not correctly formatted (appearing by author name) such as the recent Pereira et al. Yy1 paper, which also does not appear in the list of references at the end of the manuscript.

Comments on the Quality of English Language

·       A very minor note on: on a couple of occasions the authors write “underwent to”, which should only be “underwent”.

Reviewer 2 Report

Comments and Suggestions for Authors

The manuscript by Puglisi et al. compared two viral vector systems (Mo-MLVs and AAVs) for the expression of the phosphorylation-resistant form of Neurogenin2, a neurogenic factor that might be used to generate neurons from glial cells in the adult brain.

The manuscript is interesting and provides novel data that is valuable for the scientific community. The experimental design and methodological approach are appropriate and described with sufficient detail. The figures, schemes, and tables (including the supplementary materials) provide relevant support information. The discussion is comprehensive and addresses all relevant results. The conclusions are supported by the results. Overall, the manuscript is sound and does not present significant issues. The manuscript has minor issues listed below that the authors can quickly address.

Minor:
-line 55: “For this...”; “this” is in the previous paragraph; what does it refer to? Clarify
-line 62: “…ever, this turned out not to be the case, …”; unnecessary, remove.
- line 67: “PTBP1”, not referred to in any other manuscript point; indicate gene full name.
- line 81 - Pereira et al., 2024; it does not appear in the reference list
- line 108 - “have been”; should be “were”
- line 120 - “Gascon 2016. “;  should be “Gascon et al 2016.”; also, insert the respective number from the reference list
- Table S1 - why is the volume provided instead of titer?; provide titer for Mo-MLVs;   
- line 167 - “have been incubated”; should be “were incubated”
- line 206 - “has been”; should be “was carried”
- typos, mostly in figure legends, should be verified.
